# A Product-Design-Change-Based Recovery Control Algorithm for Supply Chain Disruption Problem

Jingze Chen, Haodong Kang and Hongfeng Wang *

College of Information Science and Engineering, Northeastern University, Shenyang 110819, China;
2110328@stu.neu.edu.cn (J.C.); 2070667@stu.neu.edu.cn (H.K.)
* Correspondence: hfwang@mail.neu.edu.cn

**Abstract:** In very recent years, large-scale disruptions brought by major global and local emergencies have posed many challenges with respect to the recovery control of supply chain systems. This work investigates a problem regarding the optimal control of a supply chain by considering product design change in order to enable manufacturers to recover their disrupted supply chain quickly. A two-layer optimization model is developed, in which the lower model is used to optimize the product design change path, and the upper model is used to select the appropriate alternative suppliers and schedule the delivery of customer orders. To solve the developed model, a hybrid ant colony optimization (HACO) algorithm is designed, which is combined with a Gurobi solver and uses some special strategies. The validity of the proposed algorithm is illustrated experimentally through computational tests and systematic comparison with the existing methods. It is reported that the losses caused by supply chain disruptions are reduced significantly. The proposed model and algorithm can provide a potentially useful tool that can help manufacturers decide upon the optimal form of recovery control when a supply chain system experiences a massive supply disruption.

**Keywords:** supply chain system; disruption recovery; recovery control strategy; product design change; ant colony optimization

## 1. Introduction

In recent years, large-scale supply chain disruptions have been caused by natural and man-made disasters, such as the 2008 U.S. subprime mortgage crisis, the 2011 Japan tsunami, the Russia–Ukraine conflict, and the COVID-19 pandemic. Once such emergencies occur, supply chain systems are faced with many problems, such as supply disruption, production disturbance, and demand fluctuation. Manufacturing enterprises face great uncertainties due to their own characteristics, and any blockage or disruption in supply chains can threaten their operational security. As an interdependent and interconnected whole, local disruptions can propagate through the supply chain and cause damage to the entire supply chain [1]. In today's uncertain economic environment, the success of enterprises largely depends on making effective strategic decisions while threatened by various disruption risks [2]. Therefore, it is important to develop effective disruption recovery control strategies for supply chain systems.

Over the past decades, many studies on supply chain system disruption recovery controls have been conducted to address disruption risks. The literature on mitigation strategies can be divided into two main aspects. One aspect is the proactive mitigation strategy for reducing supply chain losses through multiple sourcing, backup suppliers, backup facilities, inventory management, and capacity buffers [3,4]. These sub-strategies can help reduce and preempt the risk of disruption to ensure the normal operation of a supply chain. Another aspect is the reactive recovery strategy, which helps a supply chain recover from disruptions quickly through the development of appropriate recovery strategies [5,6]. These recovery strategies mainly include redirecting material flows, reallocating resources

and changing manufacturing plans, and reconfiguring a supply chain by rematching demand and supply points. Some scholars have combined proactive and reactive strategies to address possible disruptions in a supply chain. Hosseini et al. [7] proposed a stochastic bi-objective MIP model to illustrate how and when to employ proactive and reactive policy decisions in supplier selection and order allocation.

With the further development of globalization along with specialization and concentration in the manufacturing industry, disruptions can affect nearly the entire global supply chain [8]. Although the existing research can help to improve the ability of the supply chain system to respond to local disruptions, it is difficult to only rely on the supply chain system itself to restore the normal operation of the entire system when emergencies cause long-term or severe disruption of the supply chain. Therefore, manufacturing enterprises need to consider how to respond effectively to the risks posed to the supply chain system by major emergencies. Lu et al. [9] considered a supply chain with two downward substitutable products and used product substitution and dual-sourcing strategies to mitigate random supply chain disruptions. Chen et al. [10] adopted a combined recovery strategy of emergency procurement and product changes to cope with the massive supply disruptions caused by the pandemic. In contrast with previous studies, the product change strategy is introduced in the disruption recovery control. The supply network is reconfigured through design changes to disrupted components. From the perspective of product design, developing disruption mitigation strategies can increase the resilience of supply chain systems. When implementing a product design change, manufacturers must consider the impact of the product change on other components [11]. In addition, product design changes can affect the stability of the supply chain and increase risk [12]. How to select a design change option to implement after disruptions to enhance the supply chain system's resilience is an important issue. The study of change propagation paths aids decision making with regard to options and reducing the risk and cost of design changes [13].

For a complex product manufacturing supply chain with multiple echelons, the manufacturer's production capacity can be affected by various disruptions, and supply disruptions from upstream suppliers can have a significant impact on the manufacturer. This paper examines the issue of disruption recovery control in manufacturing supply chains in the event of a large-scale supply disruption caused by an emergency. To alleviate the losses suffered by manufacturing enterprises after disruptions, a two-layer optimization model is developed considering product design change and substitute product strategies. The model is an extension of the disruption recovery model proposed by Chen et al. [10]. The lower model makes decisions about the change path and the level of change for the product module, with the objective of minimizing the cost of change. The upper model makes decisions regarding the sourcing options of alternative suppliers after the product design change, with the objective of maximizing the total profit of the supply chain. The objective value of the upper model needs to be obtained through the optimization of the lower model, which, in turn, influences the decisions of the upper model. As the developed model was NP-hard, it is necessary to develop an efficient method to solve the developed model. While solving NP-hard problems using exact methods requires significant computational efforts, metaheuristic algorithms provide near-optimal solutions in a reasonable time frame. Therefore, a hybrid ant colony optimization (HACO) method incorporating Gurobi solver and some particular strategies has been developed to solve the model with reduced computational complexity. Therefore, the main objectives of this study are to:

- Develop a two-layer model that considers product design change options and the abilities of alternative suppliers;
- Develop a computationally efficient algorithm to solve the mathematical model without sacrificing the optimal objective function value;
- Analyze the impact of the proposed decisions on the total profit of the supply chain after disruptions.

The rest of the paper is organized as follows: Section 2 presents a literature review. In Section 3, the problem statement and the underlying assumptions are given. The

mathematical model for the problem is formulated in Section 4. The solution methodologies are described in Section 5. In Section 6, the results are presented along with a comparison of the performance levels of the proposed algorithms. Section 7 provides our conclusions and recommendations for future research.

## 2. Literature Review

This section first reviews the research on supply chain system disruption recovery and product design changes. Then, the knowledge gap between the model addressed in our work and the existing literature is introduced.

### 2.1. Supply Chain System Disruption Recovery Control Strategies

According to our analysis of the existing literature, there are two strategies for dealing with the disruptions, i.e., preventive mitigation strategies implemented before disruptions and recovery strategies implemented after disruptions, respectively [6]. Regarding the preventive mitigation strategies, the most common approaches are multiple sourcing and the use of backup suppliers. Namdar et al. [14] investigated the use of sourcing strategies including multiple sourcing, backup supplier contracts, and spot purchasing with respect to achieving supply chain resilience in the event of a disruption. Huang et al. [15] proposed a dual sourcing model that reserves capacity for emergency purchases so that emergency orders can be considered within a given capacity constraint after disruptions. Bottani et al. [16] proposed a bi-objective mixed-integer programming model that uses multiple sourcing policies to cope with unexpected fluctuations in market demand and disruptions in raw material supply and designed an ACO algorithm to solve the model. Moreover, some scholars addressed the risk of disruption from the perspective of safety inventories and risk mitigation inventories. Simchi-Levi et al. [17] coordinated process flexibility and product inventory strategies to mitigate possible supply chain disruption risks and established a two-stage robust optimization model. Lucker et al. [18] studied inventory and reserve capacity to mitigate supply chain disruption risk under the influence of stochastic demand and characterized four main risk mitigation strategies. Lucker et al. [19] optimized the location and quantity of risk mitigation inventories and reserve capacity in a serial multi-stage supply chain to cope with the disruption risk at each stage. Shahed et al. [20] developed a three-stage supply chain disruption mitigation model to address possible supplier and retailer disruptions. This study proposed an inventory policy using the renewal reward theory to maximize the profits of manufacturers after disruptions.

Another aspect is the disruption recovery strategies for allowing a supply chain to recover quickly from disruptions. Many scholars have proposed disruption recovery strategies based on the type and extent of the disruption [21]. Paul et al. [22] proposed a supply disruption recovery method for a three-stage supply chain system and developed a heuristic method. The authors further extended the mathematical model and heuristic method, developing them into a dynamic method for addressing multiple supply disruptions in real time. Paul et al. [23] developed a production recovery model for a three-stage supply chain considering three types of sudden disturbances and proposed a heuristic to obtain a recovery plan for each disturbance type for a finite future period after the disturbance. Malik and Sarkar [24] developed a recovery model that considers the major costs of the production system as well as budget and storage space constraints in order to obtain the optimal manufacturing lot size for multiple materials within the recovery time window. Paul et al. [25] developed a production recovery model for high-demand products during COVID-19 to maximize manufacturer profits by changing production plans, considering the increased demand, limited production capacity, and supply disruption. In addition, product substitution is another commonly used recovery strategy [26]. Khalilabadi et al. [27] constructed a multi-stage stochastic integer programming model using production substitution strategies to address shortages caused by production disruptions and proposed a progressive hedging algorithm to solve the model. Saha et al. [28] developed a nonlinear integer programming model to minimize the total cost of the supply chain by considering substitute products

and backorders and designed an improved particle swarm optimization algorithm to solve the model. After analyzing the existing literature, it is evident that disruption recovery control strategies designed from the supply chain system itself are relatively mature, but few consider recovery strategies from a product design perspective in order to cope with large-scale supply disruptions.

### 2.2. Product Design Change

After supply disruptions, manufacturers can adopt product substitution strategies to ensure an order's delivery. In the process of product substitution, whether the change is a partial change in the product or the design of a complete variant, it is necessary to rely on the selection of product design change options [29]. The propagation path of component changes and the planning of raw materials should be considered in the product design change process [30]. A number of scholars have developed optimal change path search methods in terms of product network models and change propagation mechanisms. Li and Zhao [31] carried out parallel processing of product design changes and presented an integrated approach for scheduling design changes by combining a simulation of change propagations with optimization algorithms to minimize change time. Uckun et al. [32] incorporated both component change cost and change probability into the adaptability index of design change plans to improve adaptability between different design change plans. Yu et al. [33] established a two-layer optimization model with the goal of maximizing customer satisfaction and minimizing change costs to solve complex product design change paths. Yu et al. [34] established a directed, weighted complex product network model to represent a product's structure under a given requirement and evaluate the impact of change propagation. Li et al. [35] proposed an entity connection model based on the logical relationship between propagation nodes to determine the optimal change propagation path in order to minimize the time required for a product design change. Stefansdottir and Grunow [36] considered the impact of product design changes on the supply chain and developed a two-stage stochastic mixed integer linear programming model to make decisions about product design options. Li et al. [37] considered the law of design change risk propagation and developed a design change risk propagation model based on the load capacity model. Rostami et al. [38] established a multi-objective mathematical model that integrates a supply chain and product development, enabling production systems to quickly respond to changes in product design and demand. The research on product design paths has mainly focused on change propagation impacts and mechanisms, which are not well integrated with supply chain systems and are rarely used in the design of disruption recovery control strategies.

### 2.3. Research Gap

The literature review has shown that the current research on supply chain disruption recovery control strategies focuses mainly on proactive defense and reactive defense strategies. Although the existing research can help improve the ability of the supply chain system to respond to local disruptions, it is difficult to solely rely on the supply chain system itself to restore the normal operation of the entire system when emergencies cause long-term or severe disruptions of the supply chain. This restoration requires manufacturing companies to consider both their own products and supply chains to develop more effective disruption recovery control strategies. In the existing research, product design change strategies are widely used by manufacturing companies to respond to changes in demand. By optimizing the change path for product components, companies can reduce the cost of design changes while maintaining product performance. However, few studies have applied product design change strategies to the recovery of manufacturing system supply chains with large-scale supply disruptions. Therefore, this study aims to investigate the disruption recovery control strategy of manufacturing supply chain systems from the perspective of collaborative decision making regarding product design and supply chain to improve the resilience of the system toward emergencies. To clearly define the problem

considered, this work builds a two-layer optimization model. Then, a HACO is introduced to deal with it. Subsequently, comparison experiments are conducted verify the validity of the HACO.

## 3. Problem Statement

In this section, the problem is defined, and the assumptions related to it are presented, thereby showcasing the main motivation for this research and the factors considered in the model.

### 3.1. Problem Definition

This paper considers a three-stage supply chain system consisting of multiple suppliers, a manufacturer, and multiple customers, as shown in Figure 1. The manufacturer produces a product for which a key raw material is supplied by multiple suppliers. Products are delivered directly from the manufacturer to the customer within the lead time, and each customer's order is independent of another order. When some or all of a manufacturing company's original suppliers are disrupted, the lack of raw material will affect the normal production of the manufacturing company, resulting in the risk of product shortages and the inability to meet downstream demand-side orders. If the original product cannot be delivered on time due to supply disruptions, the manufacturer can adopt the strategy of delayed delivery on the one hand to compensate the customer whose product cannot be delivered on time. On the other hand, the manufacturer can adopt the product design changes to produce substitute products for delivery to the customer. The manufacturer needs to compensate for delivering substitute products to customers. The main objective of this study is to maximize the total profit of a supply chain after supply disruptions.

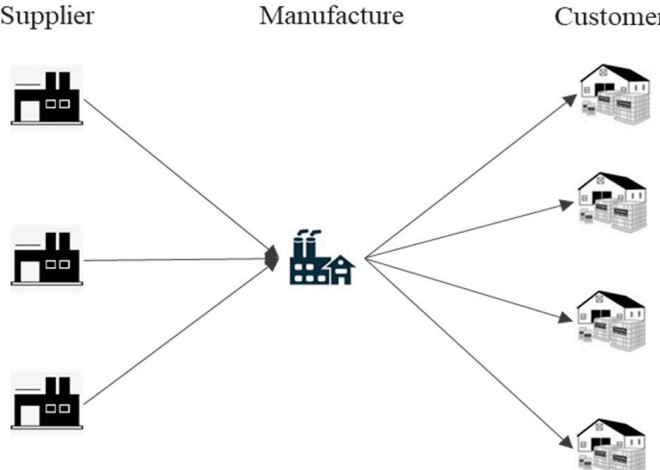

**Figure 1.** Three-stage supply chain system.

Considering the scale of supply disruptions, whether the manufacturer will adopt a product design change strategy must be determined first. If the manufacturer does not adopt product design changes, only emergency sourcing strategies will be considered; otherwise, decisions on product design change plans need to be made. When selecting a product design change solution, the key raw material is used as the starting point for the change. Then, the impact intensity of changes between configurable units is evaluated as the edge weight of the network, and the change impact analysis network is determined. Finally, with the goal of minimizing the impact of changes, the optimal change path is determined by considering the change time and cost and the abilities of alternative suppliers. An example of a product change path is shown in Figure 2, where gray represents the selected initial change node and red represents the node through which the change path passes. The selected initial change node, the connections, and the change impact between the nodes are first given on the left; then, the right shows that the change path with the minimum

change impact is obtained subject to the initial change impact intensity of the selected initial change node being satisfied. Original and substitute products will be produced simultaneously. After considering customer requirements, delivery times, and backorder losses, the order delivery sequence is determined. If no products are delivered to a customer by the deadline, the demand remains unmet. The total unmet demand after the planning horizon is considered a lost sale in the supply chain.

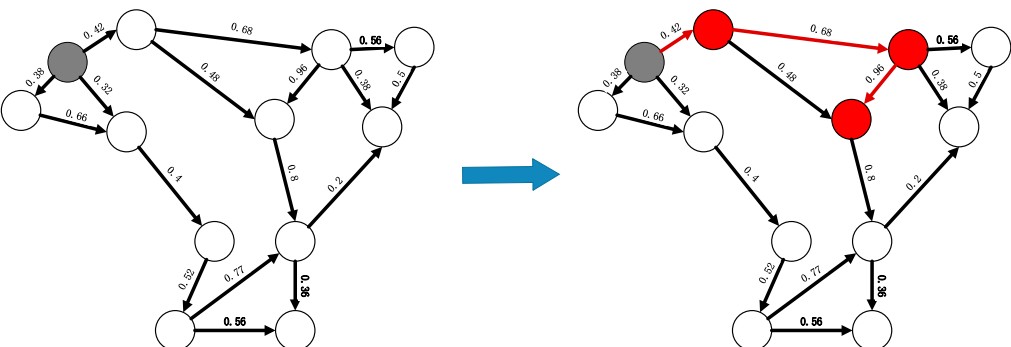

**Figure 2.** An example of a product design change path.

*3.2. Assumptions*

A number of assumptions have been made to formulate the problem, as follows.

- All products require a core component, and only the supply disruption and impact of the component are considered. Product design changes adopt this component as the starting point for the change path.
- Each customer's demand for products is constant, and customer order requirements are independent of each other.
- Whether a supplier will be disrupted is independent of the circumstances of the other suppliers. Supply disruptions occur at zero, after which point no new disruptions occur during the production planning, and the disrupted supplier cannot resume the production plan.
- Orders from customer *j* are produced in one cycle and shipped immediately after production, and products produced in cycle *t* are delivered in cycle *t* + 1, regardless of product inventory.
- The manufacturers choose only one design change option to produce substitute products to deal with disruptions.
- Substitute products and backorders are available to all customers. The manufacturer needs to compensate for delivering substitute products to customers.
- Exceeding the customer's latest delivery period $T_j$ will result in backorder costs, while exceeding the customer's latest cancellation period $U_j$ will result in lost sales costs.
- The emergency sourcing of raw materials is implemented immediately after supply disruptions, regardless of delays in production caused by emergency sourcing.
- The transportation time and cost of raw materials and products between suppliers, manufacturers, and retailers are not considered.

## 4. Mathematical Model

*Model Development*

In this section, a two-layer optimization model for product design change path planning and alternative supplier selection is developed. The upper model is related to the lower model by the necessary constraints. The lower model makes decisions about the change path for the product module with the objective of minimizing the cost of a change. In product change path planning, the intensity of the change impact between nodes and the capacity of alternative suppliers after the change must be considered. The intensity of the change impact is evaluated with reference to five indicators: functional relevance between nodes, the node change propagation coefficient, the node change propagation

probability, change time, and change cost. We first give a mathematical expression for each of the different indicators.

Functional relevance $R_{mn}$ denotes the association between node $m$ and node $n$. We assume that it contains five kinds of relevance relationships, which can be represented as follows:

$$R_{mn} = \begin{cases} 1, & \text{The two nodes need to be used simultaneously;} \\ 0.8, & \text{The two nodes are strongly associated;} \\ 0.6, & \text{The two nodes have a certain correlation;} \\ 0.2, & \text{The two nodes are weakly associated;} \\ 0, & \text{The two nodes are independent of each other.} \end{cases} \tag{1}$$

The node change propagation coefficient $D_m$ is related to the node's in-degree $D_m^{in}$ and out-degree $D_m^{out}$, which can be expressed as follows:

$$D_m = \frac{D_m^{out} - D_m^{in}}{D_m^{out} + D_m^{in}}, \; D_m \in [-1, 1] \tag{2}$$

Given the probability $P_m$ of the change of each node and the probability $P_{m \cap n}$ of the change of the current node and its neighboring nodes simultaneously, the change propagation probability $P_{mn}$ of the node can be calculated as follows:

$$P_{mn} = \frac{P_{m \cap n}}{P_m} \tag{3}$$

In summary, the intensity of the change impact $I_{mn}$ can be expressed as follows:

$$I_{mn} = \begin{cases} \frac{1 - P_{mn}}{R_{mn}} (\omega_1 D_m + \omega_2 Tc_m + \omega_3 Cc_m), & R_{mn} \neq 0, \omega_1 + \omega_2 + \omega_3 = 1 \\ 0, & R_{mn} = 0 \end{cases} \tag{4}$$

We have developed mathematical expressions for all of the indicators under consideration. Therefore, the objective function is to minimize the total intensity of the product change impact (PCI), where the reliability of the initial nodes of the design change option is considered as follows:

$$min \; PCI = \sum_{l \in L} z_l (1 - r_l) * \sum_{m \in N} \sum_{n \in N} I_{mn} x_{mn} \tag{5}$$

The model is subject to the following constraints, as presented in Equations (6)–(12). s.t.

$$\sum_{m \in N} \Delta S_m y_m \geq \sum_{l \in L} S_l \tag{6}$$

$$\sum_{m \in N} y_m \leq N \tag{7}$$

$$\sum_{m \in N} \sum_{n \in N} x_{mn} \leq 2N \tag{8}$$

$$x_{mn} \leq e_{mn}, \; \forall m, n \in N \tag{9}$$

$$\frac{\sum_{m \in N} \sum_{n \in N} \sum_{l \in L} x_{mn} z_l}{2} + 1 \geq \sum_{m \in N} y_m \tag{10}$$

$$\sum_{l \in L} z_l = 1 \tag{11}$$

$$x_{mn}, y_m, z_l \in \{0, 1\}, \; \forall m, n \in N, \; l \in L \tag{12}$$

Equation (6) ensures that the intensity of the initial change impact is consumed. Equations (7) and (8) constrain the relationship between nodes and edges in the change path. Equation (9) ensures that only edges that are connected between nodes can be selected. Equation (10) ensures that all nodes in the change path are connected. Equation (11) imposes a limitation ensuring the selection of only one initial change node, i.e., only one starting node per change path can be selected. Equation (12) constrains the binary nature of the decision variables $x_{mn}$, $y_m$, and $z_l$.

According to the determined product design change path, the product design change cost $h$ and product change time $b$ can be calculated as follows:

$$h = \sum_{m \in N} y_m C c_m \tag{13}$$

$$b = \sum_{m \in N} y_m T c_m \tag{14}$$

The manufacturer's revenue can be calculated as follows:

$$Rev = \sum_{k \in K} \sum_{t \in T} w_{kt} d_k Re \tag{15}$$

The manufacturer's total cost (*TC*), which includes the original supplier's procurement cost, the product change cost, production cost, backorder cost, and lost sales cost, can be calculated as follows:

$$
\begin{aligned}
TC = &\sum_{t \in T} \sum_{i \in I} C_i Y_{ti} + \sum_{t \in T} \sum_{i \in I} E_i (Y_{ti} - X_{ti} u_i) + \sum_{t \in T: t \geq b} \sum_{l \in L} \sum_{j \in J} Z_{tlj} (Ce_j + \\
&h) + \sum_{t \in T} Pc \left( \sum_{i \in I} Y_{ti} + \sum_{l \in L} \sum_{j \in J} Z_{tlj} \right) + \sum_{t \in T} \sum_{l \in L} \sum_{j \in J} Z_{tlj} Uc + \\
&\sum_{k \in K} B_k d_k \left( \sum_{t \in T} w_{kt} - \sum_{t \in T: t \leq T_k - 1} w_{kt} \right) + \sum_{k \in K} L_k d_k \left( 1 - \sum_{t \in T: t \leq U_k - 1} w_{kt} \right)
\end{aligned} \tag{16}
$$

The upper model makes decisions on alternative supplier sourcing plans and customer order delivery options with the objective of maximizing the manufacturer's total profit (*TP*).

$$\text{Min } TP = REV - TC \tag{17}$$

The model is subject to the following constraints, as presented in Equations (18)–(26): s.t.

$$Z_{tlj} \leq z_l Qn_{tlj}, \ \forall t \in T, \ l \in L, \ j \in J \tag{18}$$

$$Y_{ti} \leq Qm_{ti}, \ \forall t \in T, \ i \in I \tag{19}$$

$$Y_{ti} \geq u_i X_{ti}, \ \forall t \in T, \ i \in I \tag{20}$$

$$\sum_{t \in T} \sum_{i \in I} Y_{ti} + \sum_{t \in T} \sum_{l \in L} \sum_{j \in J} Z_{tlj} \leq \sum_{t \in T} Qp_t \tag{21}$$

$$In_{t-1} + \sum_{i \in I} Y_{ti} + \sum_{j \in J} Z_{tJ} - In_t = \sum_{k \in K} d_k w_{kt}, \ \forall t \in T \tag{22}$$

$$\sum_{t \in T} w_{kt} \leq 1, \ \forall k \in K \tag{23}$$

$$\sum_{t' \in T: t' \leq t} \sum_{k \in K} d_k w_{kt} \leq \sum_{t' \in T: t' \leq t-1} \left( \sum_{i \in I} Y_{ti} + \sum_{l \in L} \sum_{j \in J} Z_{tlj} \right) \tag{24}$$

$$w_{kt} \in \{0,1\}, \ \forall t \in T, \ k \in K \tag{25}$$

$$Y_{ti}, Z_{tlj} \ are \ positive \ integers, \ \forall t \in T, \ i \in I, l \in L, \ j \in J \tag{26}$$

Equation (18) constrains the supply capacity of alternative suppliers after a product change. Equations (19) and (20) constrain the supply capacity of original suppliers after a disruption. Equation (21) ensures that the maximum procurement quantity does not exceed the manufacturer's production capacity. Equation (22) balances the manufacturer's procurement, production, and inventories with the order requirements for each period. Equations (23) and (24) ensure that each customer's order can only be placed once and that the quantity of products for all orders does not exceed the manufacturer's raw material purchases during the production cycle. Equation (25) constrains the binary nature of the decision variable $w_{kt}$. Equation (26) defines the decision variables $Y_{ti}$ and $Z_{tlj}$ as positive integers.

## 5. Solution Approach

The upper model in the two-layer optimization model developed in this paper is a mixed-integer linear programming model, while the lower model is a non-linear programming model and is NP-hard. There is coupling between the two layers of the model that needs to be optimized simultaneously. The ant colony algorithm is a meta-heuristic algorithm proposed by Dorigo et al. that uses the positive feedback mechanism of the algorithm to obtain the optimal solution [39]. The ants select paths based on the concentration of pheromones left on the path according to a set probability formula. The process of parallel searching at multiple points and the use of heuristic factors grants the ant colony algorithm high solving efficiency and convergence speed. The ant colony algorithm has been widely used in path optimization related to product design change propagation [13]. Gurobi is widely used as an efficient solver of linear programming problems. Therefore, we developed a HACO algorithm in conjunction with Gurobi to solve our mathematical problem, the details of which will be presented in this section.

### 5.1. Solution Construction

To solve the supply disruption recovery problem, we first need to decide upon the product design change plan, i.e., the component change path. The number corresponding to the change node is used as an element of the ant colony contraindication table, and the paths in the contraindication table that meet the constraints of the model are a set of feasible solutions. For instance, the solution (1, 2, 3, 8, 9, 11) indicates that components are changed as a sequence with a change impact intensity of 2.9214, as shown in Figure 3. Each ant starts from the initial change node, i.e., the node numbered 1. The edge weight between nodes is the change impact intensity. The ant will choose a path based on the pheromone matrix and the edge weights during the seeking process until the model constraints are satisfied and the ant stops moving.

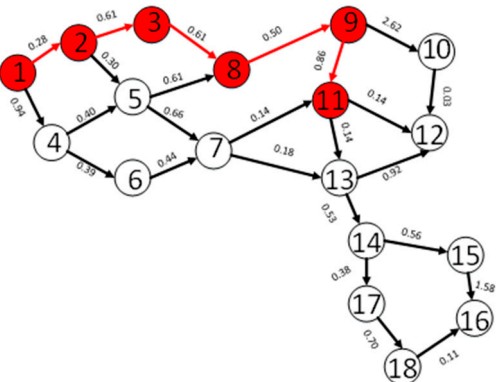

**Figure 3.** An example of a solution.

To address the characteristics of the product design change problem, when we select a node to be changed, we will encounter the constraint that certain nodes cannot be selected when we select the next change node. To address this situation, we will make a judgement on $I_{mn}$, and when $I_{mn}$ is not 0, this means that the node can be selected. Therefore, these

selectable nodes are constructed as a list and then selected from the constructed list. In addition, nodes that have already been selected cannot be repeatedly selected, and we use a taboo table to achieve this constraint. Each ant has its own separate taboo table, and for each node it passes, that node is added to the taboo table to ensure that it is not selected repeatedly.

### 5.2. Path Selection Process

The algorithm adopts a roulette strategy to select nodes for the ant's forward path. The pheromone concentration between nodes *m* and *n* in the change path decreases gradually with the number of iterations. The prediction of the ant's path from node *m* to *n* corresponds to the inverse of the change propagation strength $I_{mn}$ of the nodes in the change propagation path. The pheromone concentration between nodes *m* and *n* and the prediction of the path together determine the movement of the ants, which, in turn, yields the probability that the next node will be selected. Individuals with higher fitness values have a higher probability of being selected, while the direct elimination of individuals with lower fitness values is still avoided, thus increasing the ability of global merit seeking to some extent.

### 5.3. Local Search Method

To address the ant colony algorithm's tendency to fall into local optima, we introduce a simulated annealing algorithm based on the crossover variation operation of a genetic algorithm to perform a local search in order to enhance the algorithm's optimality-finding ability. At the same time, the new paths obtained after the crossover and mutation operations are selected and optimized to further improve the optimization ability of the algorithm in accordance with the model and constraint characteristics of this study.

At each iteration of the ant colony algorithm, a list of size 2 is maintained to store the two change paths with the least change intensity found by the contemporary ants; then, a crossover and mutation operation is performed with the two change paths for local search. For the new paths obtained via crossover and mutation, we perform neighborhood shifts according to the following formula:

$$\Delta f = newLength - length \tag{27}$$

$$P = \begin{cases} 1, & \Delta f < 0 \\ \exp\left(\frac{-\Delta f}{T_k}\right), & \Delta f > 0 \end{cases} \tag{28}$$

where *newLength* indicates the change impact intensity of the new path obtained via crossover variation, *length* indicates the change impact intensity of the change path with a higher change impact intensity in the route, and *P* denotes the probability of moving into a different neighborhood.

### 5.4. Pheromone Update Rules

In the basic ant colony algorithm, the residual pheromone should be updated after each ant has completed a step or a solution. Based on the features of the problem studied in this work, the following improvements are made to the pheromone update rules.

#### 5.4.1. Adaptive Pheromone Volatility Factor

The volatility factor $\rho$ determines the duration for which the pheromone secreted by the ants remains on the path, and the value of $\rho$ affects the overall search capacity and efficiency of the algorithm. In this paper, we set the value of the adaptive pheromone volatility factor $\rho$ so that it changes with the number of iterations, which can be expressed as follows:

$$\rho = \begin{cases} 0.8, & 0 \leq iter < 0.4Nc \\ 0.6, & 0.4Nc \leq iter < 0.6Nc \\ 0.5, & 0.6Nc \leq iter < Nc \end{cases} \tag{29}$$

where *iter* denotes the current number of iterations and *Nc* denotes the total number of iterations.

### 5.4.2. Elitist Ant System

In this paper, we have chosen to use the Gurobi solver to determine the maximum profit in the upper model. Substituting the change path generated by each ant into the upper model to solve each iteration would incur an unacceptable computational cost. Therefore, this paper adopts the idea of an elite ant system, where only the path with the least intensity of change impact in each iteration, i.e., the optimal change path, is passed to the upper model, which is then solved using Gurobi. This strategy can concentrate the search behavior of the ants around the optimal solution and improve the quality and convergence speed of the solution. The improved pheromone update formulas are as follows.

$$\tau_{mn}(t+n) = (1-\rho) * \tau_{mn}(t) + pr * \Delta\tau_{mn}(t) \tag{30}$$

$$\Delta\tau_{mn}(t) = \begin{cases} \frac{Q}{I_{mn}}, & (m,n) \in route \\ 0, & (m,n) \notin route \end{cases} \tag{31}$$

where *pr* is the maximum profit determined by the current change scheme, $\tau_{mn}(t)$ denotes the amount of pheromone left in path $(m, n)$ in each iteration, and $\Delta\tau_{mn}(t)$ denotes the incremental amount of information in path $(m, n)$ in this cycle.

### 5.4.3. Maximum and Minimum Pheromone Strategy

To avoid premature convergence of the algorithm and thus premature maturity, the pheromone on each path is restricted to the interval $[\tau_{min}, \tau_{max}]$. The pheromone content of each path is limited to the upper and lower limits of the allowed pheromone content, i.e., $\tau_{mn}(t) = \tau_{min}$ if $\tau_{mn}(t) < \tau_{min}$, and $\tau_{mn}(t) = \tau_{max}$ if $\tau_{mn}(t) > \tau_{max}$. This prevents the pheromone content of some paths from being much higher than that of other paths and causing the ants to concentrate excessively on the same path, thus enhancing the global search capacity of the algorithm to a certain extent.

After the discussion on all the components of the HACO, its pseudo-code is provided in Algorithm 1.

---

**Algorithm 1:** HACO.

---

**Input:** Initial point change intensity ($S_l$), point change cost ($Cc_m$), point change time ($Tc_m$), change impact intensity ($I_{mn}$), and a termination condition.
**Output:** The obtained best solution.
**Begin**
  **for** each edge in *pheromoneGraph* **do**
    Set initial pheromone.
  **end for**
  **for** each iteration **do**
    **for** each initial point *l* **do**
      $S_l \rightarrow changeIntensity$
      **for** each ant **do**
        Initialize ant with starting at initial point.
        **while** *changeIntensity* > 0 **do**
          Roulette selects next point
          **if** next point is not null **then**
            Update *changeIntensity*
            Update *route, length* by $I_{mn}$
            Update ant current Point
          **else**

---

**Algorithm 1:** *Cont.*

---

        Initialize ant, *changeIntensity*, *route*, *length*
       **end if**
     **end while**
    Update *inputRoutes*, *inputLengths*
   **end for**
   Local search with inputting *inputRoutes*, *inputLengths*, *initPoint*
   Obtain the best *route* and *length* for this iteration by local search
   Calculate the change cost and time based on the best route
   Obtain *profit* by using Gurobi with the $Cc_m$ and $Tc_m$
   **for** each edge in *pheromoneGraph* **do**
     Update pheromone using the pheromone update rules
   **end for**
   **if** *profit > maxProfit* **then**
     Update *minRoute*, *minLength*, *maxProfit*, *initPoint*
   **end if**
  **end for**
 **end for**
**End.**

---

## 6. Numerical Experiments

This section verifies the feasibility of the proposed model and the validity of the proposed HACO algorithm through numerous examples. The basic ant colony algorithm (ACO) and ant colony algorithms using different strategies are used for comparison with the proposed algorithm. In addition, two optimization algorithms, i.e., a genetic algorithm (GA) and a memetic algorithm (MA), are chosen for comparison. The parameters are determined based on the assumptions made for the model, and values are assigned to each parameter through a randomly generated data set. Random experiments are conducted on a large number of test problems and the performance of the algorithms is compared statistically. All the algorithms and experiments were coded in Pycharm 2021, and the codes were executed on an Intel Core i5 processor with a 3.4 GHz CPU and 12 GB of RAM.

Due to the lack of a datasets that are consistent with the research problem, we generated a random dataset *scp_i_m* for different supply chain network sizes and product design change node sizes based on the characteristics of the problem and model. In the dataset *scp_i_m*, *i* represents the number of original suppliers in the supply chain network, which, in turn, represents the size of the supply chain network, and *m* represents the number of nodes that can be selected for product design changes. For instance, scp_10_20 represents a supply chain network of 10 original suppliers that produce products with 20 selectable nodes when design changes are made. Nine such instances are examined in this work, where the number of original suppliers is $i \in \{10, 15, 20\}$ and the number of nodes available for design changes is $m \in \{20, 40, 60\}$.

### 6.1. Parameter Setting and Sensitivity Analysis

In order to analyze the influence of parameters on the performance of the HACO, several experiments with different parameter settings were carried out, and the results obtained were analyzed using the Taguchi approach. An instance with 12 original suppliers, 5 alternative suppliers, 5 initial change nodes, and 40 selectable change nodes was employed. The HACO contains four key parameters, i.e., the pheromone concentration heuristic factor $\alpha$, the expectation heuristic factor $\beta$, the pheromone volatilization factor $\rho$, and pheromone intensity $Q$. Each parameter has four levels, i.e., $\alpha = \{1, 2, 3, 4, 5\}$, $\beta = \{1, 2, 3, 4, 5\}$, $\rho = \{0.4, 0.5, 0.6, 0.7, 0.8\}$, and $Q = \{20, 30, 40, 50, 60\}$. Therefore, we select an orthogonal array $L_{25}(5^4)$ including 25 parameter combinations.

The HACO executes each parameter combination for 10 runs independently, and the average objective function value over 10 independent runs is calculated as the response

variable (RV), as can be seen in Table 1. The factor level trend of each parameter has accordingly been plotted in Figure 4.

**Table 1.** RV results of HACO.

| $\alpha$ | $\beta$ | $\rho$ | $Q$ | RV |
|---|---|---|---|---|
| 1 | 1 | 0.4 | 20 | 3,808,061 |
| 1 | 2 | 0.5 | 30 | 3,804,925 |
| 1 | 3 | 0.6 | 40 | 3,809,832 |
| 1 | 4 | 0.7 | 50 | 3,816,376 |
| 1 | 5 | 0.8 | 60 | 3,810,651 |
| 2 | 1 | 0.5 | 40 | 3,807,243 |
| 2 | 2 | 0.6 | 50 | 3,804,925 |
| 2 | 3 | 0.7 | 60 | 3,810,651 |
| 2 | 4 | 0.8 | 20 | 3,813,105 |
| 2 | 5 | 0.4 | 30 | 3,808,743 |
| 3 | 1 | 0.6 | 60 | 3,809,015 |
| 3 | 2 | 0.7 | 20 | 3,811,469 |
| 3 | 3 | 0.8 | 30 | 3,809,833 |
| 3 | 4 | 0.4 | 40 | 3,809,697 |
| 3 | 5 | 0.5 | 50 | 3,809,697 |
| 4 | 1 | 0.7 | 30 | 3,811,333 |
| 4 | 2 | 0.8 | 40 | 3,804,790 |
| 4 | 3 | 0.4 | 50 | 3,807,925 |
| 4 | 4 | 0.5 | 60 | 3,808,197 |
| 4 | 5 | 0.6 | 20 | 3,808,879 |
| 5 | 1 | 0.8 | 50 | 3,815,422 |
| 5 | 2 | 0.4 | 60 | 3,807,243 |
| 5 | 3 | 0.5 | 20 | 3,808,879 |
| 5 | 4 | 0.6 | 30 | 3,816,240 |
| 5 | 5 | 0.7 | 40 | 3,816,376 |

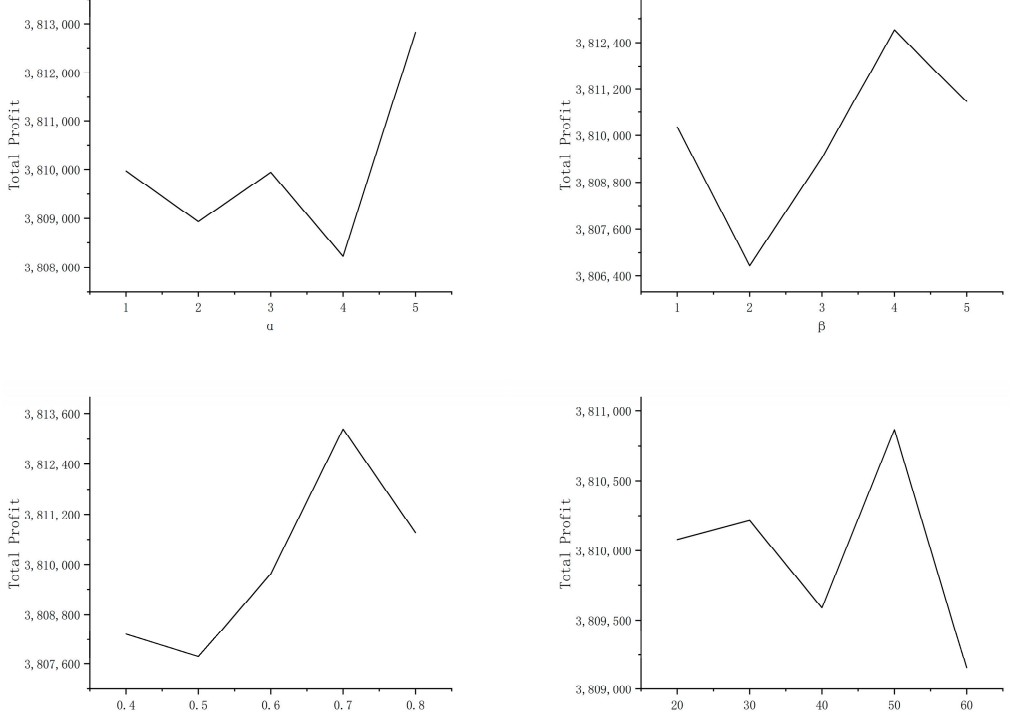

**Figure 4.** Factor level trend of HACO for each user parameter.

By analyzing the experimental results, it can be seen that $\beta$ and $\rho$ play the most important and second most important roles, respectively, as they greatly influence the HACO's exploration and exploitation abilities. Moreover, $\alpha$ and $Q$ play the third and fourth most important roles, respectively. The aforementioned results suggested the following promising parameter combination: $\alpha = 5$, $\beta = 4$, $\rho = 0.7$, and $Q = 50$; thus, these preference parameters are used in the considered experiments.

### 6.2. Comparison of Algorithms

In this section, two swarm intelligence methods, i.e., GA and MA, serving as competitive approaches, are entered into the comparison pool for the sake of fair assessment. To analyze the performance of the HACO with respect to its peers more obviously, we give the instances at various scales according to the number of original suppliers and selectable change nodes. Ten independent runs are performed for each instance for all optimization methods. In this work, we test the performance of the algorithms according to the mean and standard variance of the results after 10 runs. The comparison results between the HACO, the GA, and the MA are shown in Table 2.

**Table 2.** Comparison results between HACO, GA, and MA.

| Instance | HACO | GA | MA |
|---|---|---|---|
| scp_10_20 | $6,693,096 \pm 0$ | $6,693,096 \pm 0$ | $6,693,096 \pm 0$ |
| scp_10_40 | $6,693,998 \pm 3810$ | $6,691,328 \pm 7350$ | $6,695,629 \pm 5465$ |
| scp_10_60 | $6,693,845 \pm 8146$ | $6,653,189 \pm 10,483$ | $6,665,607 \pm 7949$ |
| scp_15_20 | $8,061,226 \pm 0$ | $8,061,226 \pm 0$ | $8,061,226 \pm 0$ |
| scp_15_40 | $8,062,048 \pm 2524$ | $8,049,030 \pm 2396$ | $8,048,161 \pm 8463$ |
| scp_15_60 | $8,054,094 \pm 3495$ | $8,027,927 \pm 2476$ | $8,042,458 \pm 8018$ |
| scp_20_20 | $9,941,884 \pm 0$ | $9,941,884 \pm 0$ | $9,941,884 \pm 0$ |
| scp_20_40 | $9,983,068 \pm 7902$ | $9,983,197 \pm 8804$ | $9,988,197 \pm 11,218$ |
| scp_20_60 | $9,957,946 \pm 6725$ | $9,821,165 \pm 18,917$ | $9,902,341 \pm 20,915$ |

By observing the optimization results obtained in Table 2, it can be found that the HACO, GA, and SA present the same performance for small-scale node instances. As the number of nodes increases, the average value of the optimal solution obtained by the HACO shows better performance than the GA and MA for all instances. Regarding the standard deviation of the optimization results, the HACO is better than the GA for all instances and better than the MA for most instances. Therefore, we can assume that the optimizer proposed in this work shows better performance compared with the GA and MA.

Furthermore, to exhibit the experimental results visually, boxplot graphs of the HACO and its competitors with different test instances are displayed in Figure 5. We can see that the results obtained by the HACO are more concentrated and stable than those acquired by its peers.

In order to validate the performance of the pheromone update rules and local search method employed in the HACO, a basic ant colony optimization (ACO) algorithm and two variants of the HACO, i.e., ACO_APF and ACO_LS, were developed. ACO_APF is designed to only adopt the improved pheromone update rules, while ACO_LS is designed to only use the improved local search method. We carried out the comparison experiments for nine instances by comparing the HACO with ACO, ACO_APF, and ACO_LS. Table 3 shows the results of the comparison.

As can be seen from Table 3, For small-scale node instances, regardless of whether the pheromone update rules and local search method are adopted, the same performance can be achieved. However, as the number of nodes increases, regarding the average value of the optimization results obtained, the HACO is significantly better than its peers for all instances. Compared to ACO and ACO_APF, ACO_LS achieved significant improvement in optimization results. The local search method can help optimizers achieve better performance. Regarding the standard deviation of the optimization results, the HACO

outperforms ACO and ACO_LS for all instances. However, the standard deviations suggest that ACO_APF can find better solutions in some instances. Nevertheless, the HACO is superior to ACO_APF since the former can find more satisfactory results. The pheromone update rules developed in this paper can enhance the stability of the algorithm and improve its search efficiency.

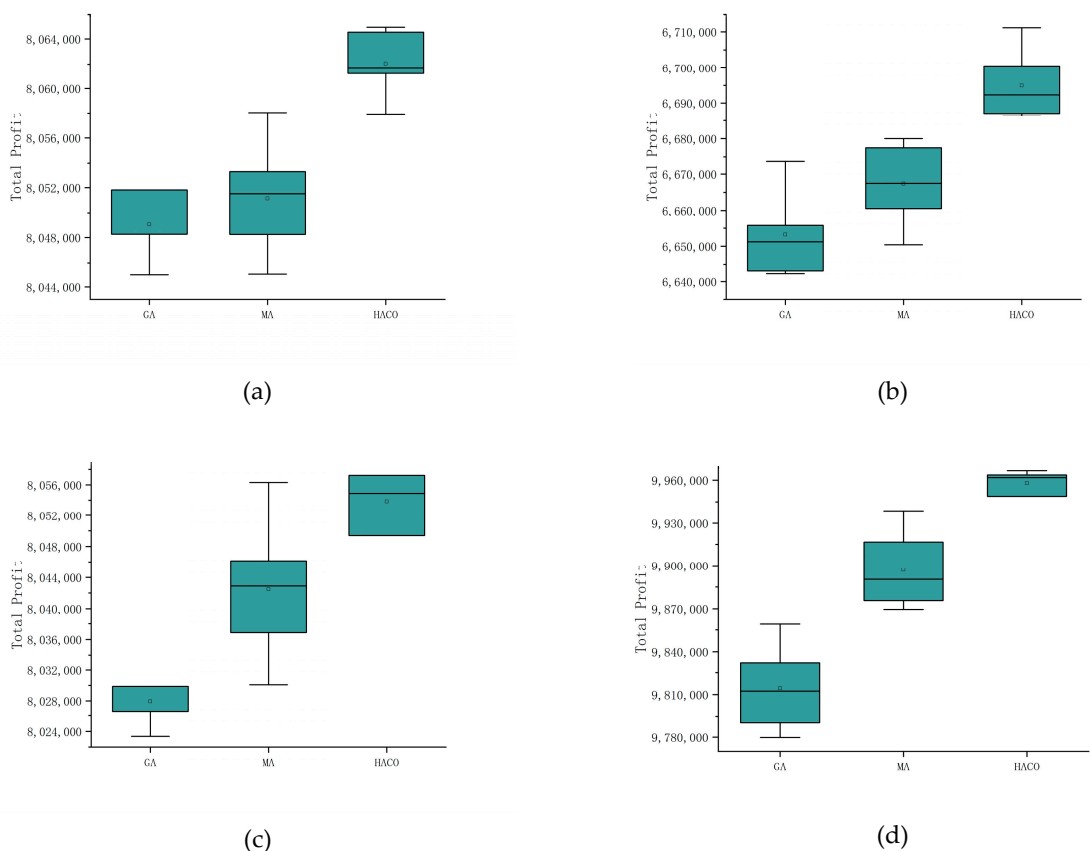

(a)

(b)

(c)

(d)

**Figure 5.** (**a**) Boxplot graphs of HACO and its competitors with respect to instance scp_15_40; (**b**) boxplot graphs of HACO and its competitors with respect to instance scp_10_60; (**c**) boxplot graphs of HACO and its competitors with respect to instance scp_15_60; (**d**) boxplot graphs of HACO and its competitors with respect to instance scp_20_50.

**Table 3.** The results of the comparison between HACO, GA, and MA.

| Instance | HACO | ACO | ACO_APF | ACO_LS |
|---|---|---|---|---|
| scp_10_20 | 6,693,096 $\pm$ 0 | 6,693,096 $\pm$ 0 | 6,693,096 $\pm$ 0 | 6,693,096 $\pm$ 0 |
| scp_10_40 | 6,693,998 $\pm$ 3810 | 6,679,116 $\pm$ 3293 | 6,691,292 $\pm$ 2016 | 6,693,193 $\pm$ 3817 |
| scp_10_60 | 6,693,845 $\pm$ 8146 | 6,642,138 $\pm$ 6863 | 6,643,189 $\pm$ 4584 | 6,657,470 $\pm$ 10,045 |
| scp_15_20 | 8,061,226 $\pm$ 0 | 8,061,226 $\pm$ 0 | 8,061,226 $\pm$ 0 | 8,061,226 $\pm$ 0 |
| scp_15_40 | 8,062,048 $\pm$ 2524 | 8,025,500 $\pm$ 7759 | 8,034,780 $\pm$ 2930 | 8,047,006 $\pm$ 6259 |
| scp_15_60 | 8,054,094 $\pm$ 3495 | 7,957,927 $\pm$ 8632 | 8,001,549 $\pm$ 2376 | 8,048,161 $\pm$ 9463 |
| scp_20_20 | 9,941,884 $\pm$ 0 | 9,941,884 $\pm$ 0 | 9,941,884 $\pm$ 0 | 9,941,884 $\pm$ 0 |
| scp_20_40 | 9,983,068 $\pm$ 6902 | 9,905,811 $\pm$ 3264 | 9,933,217 $\pm$ 1609 | 9,975,491 $\pm$ 11,211 |
| scp_20_60 | 9,957,946 $\pm$ 6725 | 9,811,579 $\pm$ 12,405 | 9,811,852 $\pm$ 3077 | 9,878,767 $\pm$ 19,978 |

To exhibit the experimental results visually, Figure 6 shows boxplot graphs of the HACO and the comparison algorithm using different strategies in different instances. It can be seen that the results obtained by the HACO are more concentrated and stable than those acquired by its peers.

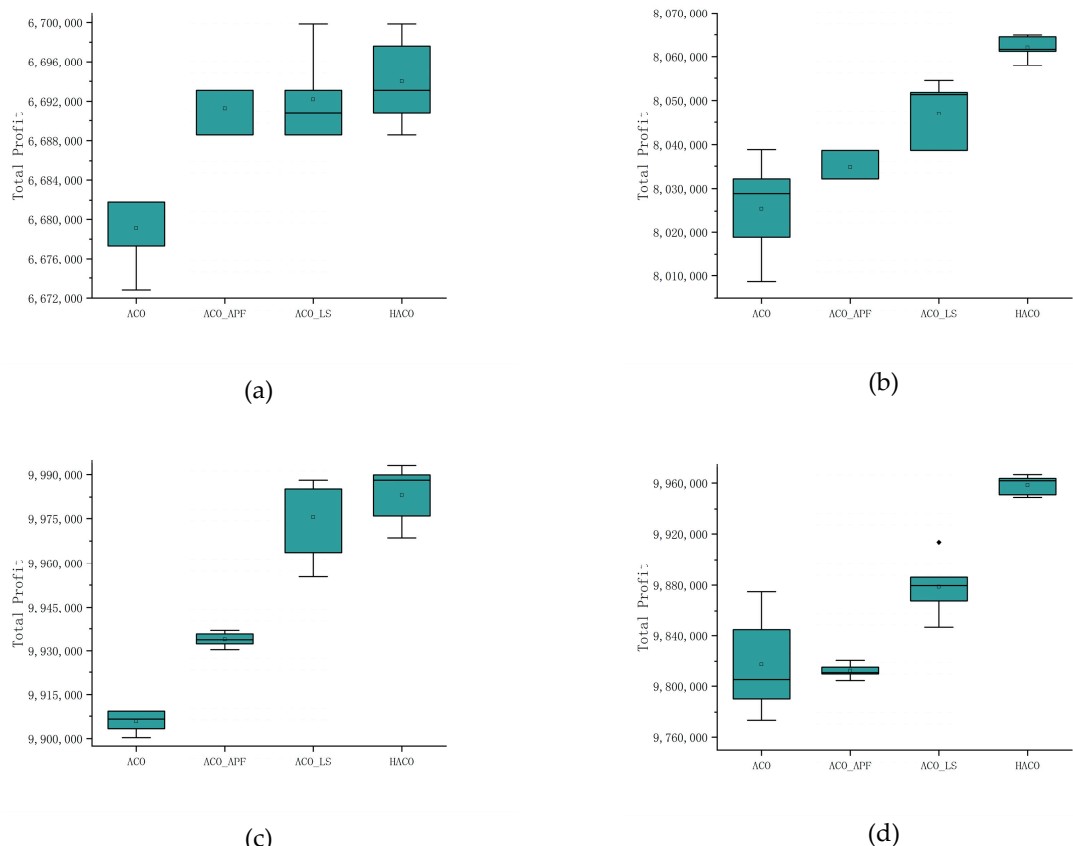

**Figure 6.** (**a**) Boxplot graphs of HACO and its peers with respect to instance scp_10_40; (**b**) boxplot graphs of HACO and its peers with respect to instance scp_15_40; (**c**) boxplot graphs of HACO and its peers with respect to instance scp_20_40; (**d**) boxplot graphs of HACO and its peers with respect to instance scp_20_60.

### 6.3. Scenario-Based Analysis

In this subsection, using the proposed HACO algorithm, the impact of employing a product design change strategy under different disruption scenarios is investigated. Accordingly, a sample problem is introduced in Table 4. If a product design change strategy is adopted after supply disruptions, $V = 1$; otherwise, $V = 0$.

**Table 4.** Parameters of sample problem.

| Parameter | Value |
|---|---|
| Number of original suppliers | 10 |
| Number of customers | 10 |
| Number of alternative suppliers | 5 |
| Number of selectable change nodes | 40 |
| Set of values concerning whether to adopt a product design change strategy ($V$) | $V = \{0, 1\}$ |

We compared the maximum total profit of the manufacturer under different disruption scenarios where a product design change strategy is adopted and in one where it is not, and the results are shown in Table 5. It can be seen that the total profit decreases as the number of disrupted suppliers increases. When the number of disrupted suppliers is small, the manufacturer can make significant profits by adopting only an emergency sourcing strategy. However, as the number of disrupted suppliers increases, the original strategy is no longer able to meet the manufacturer's production demands. However, with a product design change strategy, the total profit loss is lower, and the gap between the two

options (i.e., adopting a product design change strategy or not) increases. This suggests that a product design change strategy can reduce profit loss in the case of large-scale supply disruptions.

**Table 5.** Numerical results regarding the impact of product design change strategy on total profit.

| Number of Disruptions | Total Profit | |
|---|---|---|
| | *V* = 1 | *V* = 0 |
| 0 | 6,999,975 | 6,999,975 |
| 4 | 6,696,798 | 5,467,688 |
| 6 | 3,797,728 | 1,567,682 |
| 8 | 1,382,646 | −1,532,377 |

The order delivery statuses of the two options for manufacturers after supply disruptions are shown in Table 6. It can be seen that as the number of disrupted suppliers increases, more orders are delayed or cancelled as the manufacturer's production falls significantly. This not only reduces the manufacturer's profitability but also affects the company's reputation. Following the adoption of the developed product change strategy, the order delivery situation improved significantly. By adopting an appropriate recovery strategy after disruptions, a manufacturer can effectively improve order deliveries and significantly reduce lost profits.

**Table 6.** Numerical results regarding the impact of product design change strategy on customer orders.

| Number of Disruptions | Delivered on Schedule | | Delayed Delivery | | Undelivered | |
|---|---|---|---|---|---|---|
| | *V* = 1 | *V* = 0 | *V* = 1 | *V* = 0 | *V* = 1 | *V* = 0 |
| 0 | 10 | 10 | 0 | 0 | 0 | 0 |
| 4 | 10 | 8 | 0 | 2 | 0 | 0 |
| 6 | 6 | 3 | 4 | 3 | 0 | 4 |
| 8 | 3 | 0 | 5 | 4 | 2 | 6 |

## 7. Conclusions

This paper investigated a make-to-order manufacturing supply chain system. A supply chain system disruption recovery control problem in the context of emergencies is studied from the perspective of product design changes. A two-layer model was developed, where the lower model makes decisions about the change path and the level of change for the product module, with the objective of minimizing the cost of change, and the upper model makes decisions on sourcing options of alternative suppliers after the product design change in order to maximize the total profit of the supply chain. To handle the proposed problem, a hybrid ant colony optimization algorithm was designed by considering its features and searching operators. The numerical experiment showed that the proposed model is effective in reducing the manufacturer's losses in the case of supply chain system disruptions. In addition, to validate the performance of the considered modified algorithm, experiments on a set of randomly generated test instances were carried out. For the sake of comparison, three approaches from the literature were chosen. The attained optimization results verified that the proposed approach exhibited better performance in solving the considered problem. In conclusion, the disruption recovery control algorithm developed in this paper can effectively reduce losses in a manufacturing supply chain in the event of large-scale supply disruptions and can be used in a wide range of manufacturing industries involving product design and manufacturing. This study can also be applied in instances of product design changes induced by demand changes to help enterprises meet market demands.

There are still issues in this paper that deserve further research and discussion. Firstly, in future studies, we will consider a supply chain system with complex product structures

and a supply network closer to a real scenario and design more effective swarm intelligence approaches to handle the studied problem. Secondly, considering the impact of critical facility disruptions on supply chain systems, developing disruption recovery control strategies from a product perspective to improve system resilience is another future research direction based on this paper [40]. In addition, the impact of supply chain disruptions and demand uncertainty on recovery control strategies could be considered in future studies [41].

**Author Contributions:** Conceptualization, J.C. and H.W.; Data curation, J.C. and H.K.; Formal analysis, J.C.; Funding acquisition, H.W.; Methodology, J.C. and H.K.; Project administration, J.C. and H.W.; Software, H.K.; Supervision, H.W.; Validation, J.C. and H.K.; Visualization, H.K.; Writing—original draft, J.C.; Writing—review and editing, J.C. and H.W. All authors have read and agreed to the published version of the manuscript.

**Funding:** This work was funded in part by the National Key Research and Development Program of China under Grant no. 2020YFB1708200, the National Natural Science Foundation of China under Grant nos. 62173076 and 71671032, and the Fundamental Research Funds for the Central Universities under Grant no. N180408019.

**Data Availability Statement:** The data presented in this study are available on request from the corresponding author. The data are not publicly available due to privacy restrictions.

**Conflicts of Interest:** The authors declare no conflict of interest.

## Notations and Variables

The notations required and used in the development of the mathematical model are defined and listed as follows:

*List of indices:*

| | |
|---|---|
| $i$ | Index for original suppliers; |
| $j$ | Index for alternative suppliers; |
| $m, n$ | Index for product part nodes; |
| $l$ | Index for initial change nodes; |
| $k$ | Index for customers; |
| $t$ | Index for production periods. |

*List of decision variables:*

| | |
|---|---|
| $x_{mn}$ | 1 if changed from node $m$ to node $n$ and 0 otherwise; |
| $y_m$ | 1 if node $m$ is changed and 0 otherwise; |
| $z_l$ | 1 if the initial change node is selected and 0 otherwise; |
| $Y_{ti}$ | Quantity to be procured in $t^{th}$ period from supplier $i$ after disruptions; |
| $Z_{tlj}$ | Quantity to be procured in $t^{th}$ period from alternative supplier $j$ after product change; |
| $w_{kt}$ | 1 if $k^{th}$ customer's order is produced in $t^{th}$ period and 0 otherwise. |

*List of product change path planning model parameters:*

| | |
|---|---|
| $\omega$ | Weight coefficient; |
| N | Maximum number of nodes in the product network; |
| $I_{mn}$ | Intensity of change impact from node to node $n$; |
| $R_{mn}$ | Functional relevance of node $m$ with node $n$; |
| $D_m$ | Change propagation impact of node $m$; |
| $D_m^{in}$ | In-degree of node $m$; |
| $D_m^{out}$ | Out-degree of node $m$; |
| $P_m$ | Probability of change of node $m$; |
| $P_{m \cap n}$ | Probability of simultaneous change of node $m$ and node $n$; |
| $P_{mn}$ | Propagation probability of node $m$ to node $n$; |
| $Tc_m$ | Change time of node $m$; |
| $Cc_m$ | Change cost of node $m$; |
| $S_l$ | Initial change impact intensity of the initial change node $l$; |
| $\Delta S_m$ | Change impact intensity of node $m$ covered by change propagation path; |
| $r_l$ | Reliability coefficient of initial node $l$. |

List of supply chain disruption recovery model parameters:

$u_i$      1 if the original supplier $i$ has not been disrupted and 0 otherwise;

$X_{ti}$      Quantity to be procured in $t^{th}$ period from supplier without disruptions;

$C_i$      Unit cost of raw materials from supplier $i$;

$E_i$      Unit emergency cost of raw materials from supplier $i$;

$Ce_{lj}$      Unit cost of alternative raw materials from alternative supplier $j$;

$Qm_{ti}$      Maximum supply capacity of supplier $i$ in $t^{th}$ period;

$Qn_{tlj}$      Maximum supply capacity of alternative supplier $j$ in $t^{th}$ period;

$Qp_t$      Maximum production capacity in $t^{th}$ period;

$Pc$      Unit cost of production;

$In_t$      Inventory in period;

$d_k$      Quantity of order demand from customer $k$;

$T_k$      Delivery lead time for customer $k$;

$U_k$      Last time for customer $k$ to cancel the order;

$B_k$      Unit cost of backorder for $k^{th}$ customer's order after delayed delivery;

$L_k$      Unit cost of lost sales for $k^{th}$ customer's order after order cancellation;

$h$      Product change cost;

$b$      Product change time;

$Uc$      Unit compensation cost of substitution products;

$Re$      Unit price of the order.

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
