# Peer review of "A Product-Design-Change-Based Recovery Control Algorithm for Supply Chain Disruption Problem"

_electronics, doi:10.3390/electronics12122552_

Round 1
Reviewer 1 Report
The evaluated article presents the original results of the research carried out by the authors. From the methodological point of view, it can be considered to be well structured, balanced, bringing new findings that are verified, albeit on random numbers. The contribution of the presented research can be positively evaluated both in terms of contribution to the development of the scientific field and in terms of potential practical applicability. The research objective is appropriately set, the outputs are adequate to the stated goal. The authors work appropriately with the literature, and especially the introductory section can be considered a successful review of the state of the art.
Overall, the reviewed article can be evaluated very positively and recommended for printing, without further modifications.
Author Response
Point 1: The evaluated article presents the original results of the research carried out by the authors. From the methodological point of view, it can be considered to be well structured, balanced, bringing new findings that are verified, albeit on random numbers. The contribution of the presented research can be positively evaluated both in terms of contribution to the development of the scientific field and in terms of potential practical applicability. The research objective is appropriately set, the outputs are adequate to the stated goal. The authors work appropriately with the literature, and especially the introductory section can be considered a successful review of the state of the art. Overall, the reviewed article can be evaluated very positively and recommended for printing, without further modifications.
Response 1: Thank you very much for taking the time and effort to review our manuscript. We would also like to thank you for your interest and approval.

Reviewer 2 Report
The following questions were left unanswered in the manuscript
1. Is the algorithm universal for all industries or is it suitable for a particular industry?
2. Is this algorithm suitable for eliminating the consequences of any failures or not?
Author Response
Point 1: Is the algorithm universal for all industries or is it suitable for a particular industry?
Response 1: The disruption recovery control algorithm developed in this paper can effectively reduce losses in the manufacturing supply chain in the event of large-scale supply disruptions. The main contribution of our work is to help manufacturing companies cope with large-scale supply disruptions from a product design perspective. Therefore, the algorithm can be used in a wide range of manufacturing industries involving product design and manufacturing, such as household electrical appliance industry and automobile industry. Our research will have some reference value for the interruption recovery of supply chain systems in other non-manufacturing industries. We have added a description of the scope in which our algorithm is applicable in lines 72-78 of the introduction and lines 561-565 of the conclusion and highlighted them in red in the manuscript.
Point 2: Is this algorithm suitable for eliminating the consequences of any failures or not?
Response 2: The disruption recovery control algorithm developed in this paper is used to cope with large-scale supply disruptions in manufacturing supply chain systems. As prolonged supply disruptions can lead to raw material shortages, which in turn affect the production of manufacturing companies. By considering a product design change strategy, it can help companies to reconfigure their supply network to quickly resume production. This study is also applicable to the impact of changes in demand on supply chains, such as during the COVID-19 pandemic when many manufacturing companies changed their products to meet market demand. However, for other disruptions in the supply chain system, recovery from a product perspective is difficult to achieve and our work may not be applicable. We highlighted them in red in lines 72-78 of the introduction and lines 561-565 of the conclusion.

Reviewer 3 Report
The paper sounds quite interesting and it seems to investigate an important and present topic. Anyway, the paper cannot be practically used nor understanded by any reader as the adopted model has not been presented properly. There is no possibility to understand the "Model development" section because of the presence of multiple unexplained variables. The "Notation and decision variables" section must be revised to be comprehensive and clear at least with the "Model development" section. Just some examples:
- in Eq. 4: what is omega? What is "c_m"? A customer? In the notation it is referred as "k" and it does not depends by "m".
- Eq. 5: what is "r_lj"?
- Eq. 6: what are "deltaS_m" and "S_l"?
- Eq. 7: what is "N"?
- ... I did not go further.
In its current status, I cannot evaluate the paper nor appreciate the author contribution. I ask authors to add a notation section which comprehend the meaning of all the variables used in the paper and to put it at the very beginning of the paper itself. Further explanation in the text adjacent to equations might be useful for the reader to understand, too.
Some other concerns:
- References are not matching in the bibliography. E.g., "Bottani et al. [18]" at line 117 is reference 15 in the bibliography. Please revise all the references in this sense. The references' format in the bibliography is not coherent, also.
- The literature review section is very extended and no insights are collected at the end of it. Authors should revise the literature review section leaving only valuable papers: some references are quite old and their actual relevance should be considered. The output of the literature review is missing: after the review, I would expect a paragraph justifying the motivation of your work with respect to available research. Then you are justified to proceed with the problem statement.
- Line 205-220. You stated that the optimal change path minimizes the impact of changes. At this point it is not clear if the target node is selected before the search of the optimal path (that, with unweighted edges should be the shortest path), or you reach it once you verify edges are too impactful. I think re-arranging Figure 2 can help in this sense, maybe adding dummy weights on edges (something more similar to Figure 3)?
- Disruptions in some supply networks may be more impactful then others. In the so called critical infrastructure systems, the resilience capacities (such as recovery) become crucial. Your work would acquire particular relevance if also applicable to this field. I was wondering how you think it can be possible to re-arrange and apply your model in the case of a critical infrastructure supply network (e.g., Patriarca R., Simone F., Di Gravio G. Modelling cyber resilience in a water treatment and distribution system (2022) Reliability Engineering and System Safety, 226, art. no. 108653, DOI: 10.1016/j.ress.2022.108653 link: https://www.scopus.com/inward/record.uri?eid=2-s2.0-85132227607&doi=10.1016%2fj.ress.2022.108653&partnerID=40&md5=bf0e8781cbd78f5ab2a706e0c3432ace).
I think you should discuss this possibility in the conclusion section.
Reviewer 4 Report
The proposed manuscript is focused on the supply chain disruption recovery modelling based on meta-heuristic algorithms use. The problem is worth investigating and adheres to the scientific scope of the Journal.
The article is well written, the structure is properly designed. The main research goals and contributions are clearly justified and presented. The proposed two-layer model is properly described and the results are clearly presented and confirmed by the material (figures, tables). The literature is up-to-date.
The reviewer has one suggestion to improve the text:
the literature section presents some research results in the investigated area. However, there is no clear summary about the research or knowledge gaps identified by the authors based on this literature review presentation; there is no clear presentation how the proposed model differs from the well-known ones. This should be improved.
Author Response
Point 1: The literature section presents some research results in the investigated area. However, there is no clear summary about the research or knowledge gaps identified by the authors based on this literature review presentation; there is no clear presentation how the proposed model differs from the well-known ones. This should be improved.
Response 1: Thank you for the careful review and valuable advice. In response to your comment, we have added section 2.3 “Research gap”, which summarizes the research in the existing literature and explains how we differ from the existing research to illustrate the motivation for our research clearly. We have highlighted it in red in the manuscript as follows:
The literature review shows that current research on supply chain disruption recov-ery control strategies focuses mainly on proactive defense and reactive defense strategies. Although the existing research can help to improve the ability of the supply chain system to respond to local disruptions, it is difficult to only rely on the supply chain system itself to restore the normal operation of the entire system when emergencies cause long-term or severe disruption of the supply chain. This requires manufacturing companies to consider both their own products and supply chains to develop more effective disruption recovery control strategies. In existing research, product design change strategies are widely used by manufacturing companies to respond to changes in demand. By optimizing the change path for product components, companies can reduce the cost of design changes while maintaining product performance. However, few studies have applied product design change strategies to the recovery of manufacturing system supply chains with large-scale supply disruptions. Therefore, this study aims to investigate the disruption recovery con-trol strategy of manufacturing supply chain systems from the perspective of collaborative decision-making of product design and supply chain to improve the resilience of the sys-tem to emergencies. To clearly define the problem considered, this work builds a two-layer optimization model. Then, a HACO is introduced to deal with it. Comparison experi-ments verify the validity of HACO.

Round 2
Reviewer 3 Report
The authors addressed all the comments made by reviewers and improved the paper accordingly. In its current form, the paper is worth to be published in Electronics.